# FAILURE MODES OF MAXIMUM ENTROPY RLHF

## ABSTRACT

In this paper, we show that Simple Preference Optimization (SimPO) can be derived as Maximum Entropy Reinforcement Learning with length-normalized temperature, providing a theoretical foundation for this reference-free method. Motivated by SimPO's strong performance in offline preference optimization, we investigate whether Maximum Entropy RL can achieve similar results in online RLHF settings. Our experiments find that Maximum Entropy RL consistently exhibits overoptimization and unstable KL dynamics, even at very low learning rates. Unlike KL-constrained methods that maintain stable training, entropy regularization fails to prevent reward hacking and appears to correlate with overoptimization. Lastly, we discuss possible explanations for why SimPO succeeds in offline settings while Maximum Entropy RL struggles in online scenarios. Our findings suggest that reference-free approaches may face distinct challenges when applied to online or offline preference learning.

## 1    INTRODUCTION

Aligning AI systems with human values is widely recognized as a central challenge in modern AI (Bengio et al., 2025; Russell, 2022). The prevailing approach, Reinforcement Learning from Human Feedback (RLHF) (Christiano et al., 2023; Stiennon et al., 2022; Ziegler et al., 2020; Bai et al., 2022; Ouyang et al., 2022), typically follows a three-stage pipeline: (1) supervised fine-tuning (SFT), (2) training a reward model from preference data, and (3) optimizing the policy with reinforcement learning under KL divergence regularization to limit deviation from a reference model. While this framework has been successful, it is computationally demanding and operationally complex, requiring separate reward models, substantial human annotation, and careful hyperparameter tuning to balance reward maximization with stability.

These limitations have motivated the exploration of direct alignment algorithms (DAAs) (Rafailov et al., 2024a) that aim to simplify the pipeline by avoiding explicit reward modeling. Direct Preference Optimization (DPO) (Rafailov et al., 2024c) is one such method, reformulating preference learning as a supervised objective with an implicit KL prior, and grounding its design in KL-constrained reinforcement learning. More recently, Simple Preference Optimization (SimPO) (Meng et al., 2024) has attracted attention for achieving strong empirical results while discarding the reference model entirely. Instead, SimPO employs length-normalized log likelihood and a target margin between preferred and dispreferred responses, yielding an objective that is simple to implement yet competitive in practice.

Despite these promising results, SimPO has lacked the kind of principled theoretical framework that underpins reference-based methods like DPO. This raises several questions: What might explain SimPO's effectiveness as a reference-free approach? Can it be connected to established reinforcement learning principles? And if so, what might such a connection imply for the broader landscape of preference optimization methods?

In this work, we take a step toward answering these questions by establishing a connection between SimPO and Maximum Entropy Reinforcement Learning (Ziebart et al., 2008). We show that SimPO can be interpreted as a closed-form solution to a Maximum Entropy RL objective with length-normalized temperature scaling. This perspective provides SimPO with a theoretical grounding analogous to DPO's relationship with KL-constrained RL, while also suggesting that reference-free optimization may arise naturally from entropy regularization under certain conditions.

At the same time, this analysis raises an empirical question: if SimPO can be viewed as an offline Maximum Entropy solution, could online Maximum Entropy RL also serve as a viable alternative to KL-constrained methods in RLHF? To explore this possibility, we conducted experiments comparing Maximum Entropy RL and KL-constrained RL on the TL;DR summarization benchmark using models from the Pythia suite.

Our experiments reveal a notable asymmetry. While SimPO performs well in offline preference optimization, online Maximum Entropy RL often exhibited instability and signs of overoptimization, even at conservative learning rates. We also observed that increases in entropy tended to correlate with such instabilities, suggesting that entropy regularization may not always guard against reward hacking and, in some cases, could contribute to it. One possible explanation is that SimPO benefits from implicit stabilizing factors—such as dataset constraints and target margins—that approximate the regularization effects of a reference model, whereas these protections are absent in online Maximum Entropy RL.

Our contributions are threefold. First, we provide a theoretical interpretation of SimPO as Maximum Entropy RL with adaptive temperature scaling, situating it within established RL frameworks. Second, we present empirical evidence that while Maximum Entropy RL is effective in offline settings (through SimPO), applying it directly in online RLHF can lead to instability and overoptimization, highlighting potential limitations of entropy regularization on its own. Third, we offer insight into why SimPO appears to succeed offline despite these challenges, pointing to the role of dataset constraints and target margins in stabilizing optimization. Together, these results provide a principled perspective on SimPO while suggesting that reference-free approaches may face important limitations in online training. We hope these findings help clarify the relationship between entropy-based methods and preference optimization, and open the door for further work on identifying the regularization mechanisms needed for robust online alignment.

## 2 BACKGROUND

In this section, we review the relevant background topics, while additional related work is provided in Appendix A.

### 2.1 CANONICAL RLHF

We reiterate the standart RLHF pipeline as outlined in (Ziegler et al., 2020) and subsequent works (Stiennon et al., 2022; Bai et al., 2022; Ouyang et al., 2022). It consists of three main stages: (1) Supervised Fine-Tuning (SFT), (2) Reward Modeling, and (3) RL Optimization.

**SFT:** A pre-trained LM is fine-tuned on task-specific high-quality data via supervised learning to obtain the initial policy $\pi^{\text{SFT}}$.

**Reward Modeling:** Prompts $x$ are sampled, and $\pi^{\text{SFT}}$ generates answer pairs $(y_1, y_2)$. Human annotators indicate preferences $y_w \succ y_l \mid x$, assumed to reflect a latent reward function $r^*(x, y)$. A common approach is to model preferences with the Bradley-Terry (BT) model (Bradley & Terry, 1952):

$$p(y_1 \succ y_2 \mid x) = \frac{\exp\left(r^*(x, y_1)\right)}{\exp\left(r^*(x, y_1)\right) + \exp\left(r^*(x, y_2)\right)}. \tag{1}$$

Given a dataset $\mathcal{D} = x^{(i)}, y_w^{(i)}, y_l^{(i)}$, we learn a reward model $r_\phi$ by minimizing the binary classification loss:

$$\mathcal{L}R = -\mathbb{E}(x, y_w, y_l) \sim \mathcal{D}\left[\log \sigma\left(r_\phi(x, y_w) - r_\phi(x, y_l)\right)\right], \tag{2}$$

where $\sigma$ is the sigmoid function. In practice, $r_\phi$ is initialized from $\pi^{\text{SFT}}$ with a linear head, and reward outputs are normalized for stability.

**RL Fine-Tuning:** Finally, the policy $\pi_\theta$ is optimized using the learned reward, constrained by a KL term to stay close to the reference policy $\pi_{\text{ref}} = \pi^{\text{SFT}}$:

$$\max_{\pi_\theta} \mathbb{E}x, y \sim \pi_\theta[r_\phi(x, y)] - \beta\mathbb{D}_{\text{KL}}[\pi_\theta(y \mid x)||\pi_{\text{ref}}(y \mid x)]. \tag{3}$$

This prevents overoptimization and distributional shift. In practice, this objective is optimized with PPO (Schulman et al., 2017), using a reward defined as $r(x, y) = r_\phi(x, y) - \beta(\log \pi_\theta(y \mid x) - \log \pi_{\text{ref}}(y \mid x))$.

## 2.2 DIRECT PREFERENCE OPTIMIZATION

Direct Preference Optimization (DPO) (Rafailov et al., 2024c) has become a popular method for preference-based tuning. Unlike traditional approaches that train a separate reward model, DPO defines the reward directly in terms of the optimized policy:

$$r(x, y) = \beta \log \frac{\pi_\theta(y \mid x)}{\pi_{\text{ref}}(y \mid x)} + \beta \log Z(x), \tag{4}$$

Here, $\pi_\theta$ is the current policy, $\pi_{\text{ref}}$ is a reference (often the SFT model), and $Z(x)$ is a normalization term. DPO incorporates this reward into the Bradley-Terry (Bradley & Terry, 1952) framework, where preference probabilities are given by:$p(y_w \succ y_l \mid x) = \sigma(r(x, y_w) - r(x, y_l))$. This leads to the following objective, computed over preference triplets $(x, y\_w, y\_l)$:

$$\mathcal{L}_{\text{DPO}}(\pi_\theta; \pi_{\text{ref}}) = -\mathbb{E}_{(x,y_w,y_l)\sim\mathcal{D}} \left[ \log \sigma \left( \beta \log \frac{\pi_\theta(y_w \mid x)}{\pi_{\text{ref}}(y_w \mid x)} - \beta \log \frac{\pi_\theta(y_l \mid x)}{\pi_{\text{ref}}(y_l \mid x)} \right) \right], \tag{5}$$

By modeling preferences directly through policy ratios, DPO removes the need for an explicit reward model while remaining grounded in a probabilistic preference framework.

## 2.3 SIMPLE PREFERENCE OPTIMIZATION

Simple Preference Optimization (SimPO) (Meng et al., 2024) is a reference-free method for preference-based fine-tuning that aligns the reward used in training with the likelihood used at inference. Unlike DPO, SimPO eliminates the need for a reference policy by defining the reward as the length-normalized log-likelihood of the model output:

$$r_{\text{SimPO}}(x, y) = \frac{\beta}{|y|} \log \pi_\theta(y \mid x) = \frac{\beta}{|y|} \sum_{i=1}^{|y|} \log \pi_\theta(y_i \mid x, y_{<i}) \tag{6}$$

This formulation ensures that the reward ranking $r(x, y_w) > r(x, y_l)$ aligns with the generation-time likelihood ranking $p_\theta(y_w \mid x) > p_\theta(y_l \mid x)$, which is often violated in DPO. SimPO also introduces a target margin $\gamma > 0$ into the Bradley-Terry model to encourage separation between preferred and dispreferred responses:

$$p(y_w \succ y_l \mid x) = \sigma(r(x, y_w) - r(x, y_l) - \gamma) \tag{7}$$

This leads to the SimPO training objective:

$$\mathcal{L}_{\text{SimPO}}(\pi_\theta) = -\mathbb{E}_{(x,y_w,y_l)\sim\mathcal{D}} \left[ \log \sigma \left( \frac{\beta}{|y_w|} \log \pi_\theta(y_w \mid x) - \frac{\beta}{|y_l|} \log \pi_\theta(y_l \mid x) - \gamma \right) \right] \tag{8}$$

## 3 SIMPO IS THE MAXIMUM ENTROPY RL

SimPO is a widely used preference alignment method, appreciated for its strong empirical performance and simplicity due to its reference-free objective. However, it lacks a theoretical foundation, unlike reference-based approaches such as DPO, which is derived from a KL-constrained RL objective. Recent work (Liu et al., 2024) made the important observation that posterior probability rewards correspond to Maximum Entropy RL in their analysis of reference policies. Building on this insight, we establish the connection between this MaxEnt formulation and SimPO, showing that SimPO can be understood as Maximum Entropy RL with adaptive temperature through length normalization.

### 3.1 MAXIMUM ENTROPY RL

Maximum Entropy Reinforcement Learning (MaxEnt RL) augments the standard RL objective with an entropy term, encouraging policies that align with the soft value function (Ziebart et al., 2008; Toussaint, 2009; Rawlik et al., 2013; Fox et al., 2015; O'Donoghue et al., 2016; Abdolmaleki et al., 2018; Haarnoja et al., 2018; Mazoure et al., 2020; Han & Sung, 2021; Zhang et al., 2025). It is deeply connected to probabilistic inference (Toussaint, 2009; Rawlik et al., 2013; Levine, 2018) and supported by both stochastic inference (Ziebart, 2010; Eysenbach & Levine, 2021) and game-theoretic foundations (Grünwald & Dawid, 2004; Ziebart et al., 2010; Han & Sung, 2021; Kim & Sung, 2023). MaxEnt is often favored for promoting exploration (Haarnoja et al., 2018; Hazan et al., 2019), smoothing optimization (Ahmed et al., 2019), and enabling robust decision-making (Eysenbach & Levine, 2021).

The general form of the Maximum Entropy Reinforcement Learning (MaxEnt RL) objective can be written as

$$\pi^\star = \arg\max_\pi \ \mathbb{E}_{\tau \sim p^\pi(\tau)} \left[ \sum_{t=1}^T r(\mathbf{s_t}, \mathbf{a_t}) + \alpha \, \mathcal{H}_\pi[\mathbf{a_t} \mid \mathbf{s_t}] \right], \tag{9}$$

where $\tau = (\mathbf{s_1}, \mathbf{a_1}, \mathbf{s_2}, \mathbf{a_2}, \ldots, \mathbf{s_T}, \mathbf{a_T})$ is a trajectory sampled under policy $\pi$, and $p^\pi(\tau) = p_1(\mathbf{s_1}) \prod_{t=1}^T \pi(\mathbf{a_t} \mid \mathbf{s_t}) p(\mathbf{s_{t+1}} \mid \mathbf{s_t}, \mathbf{a_t})$ denotes the trajectory distribution induced by $\pi$. The term $\mathcal{H}_\pi[\mathbf{a_t} \mid \mathbf{s_t}] = - \int \pi(\mathbf{a_t} \mid \mathbf{s_t}) \log \pi(\mathbf{a_t} \mid \mathbf{s_t}) \, d\mathbf{a_t}$ represents the conditional entropy of the policy at each time step, and the temperature coefficient $\alpha$ controls the trade-off between reward maximization and policy stochasticity.

### 3.2 SIMPO FROM MAXIMUM ENTROPY RL

RLHF is commonly modeled as a contextual bandit problem, though some approaches treat it as a token-level MDP (Rafailov et al., 2024b; Xie et al., 2024). In this work, we adopt the contextual bandit view (Elwood et al., 2023), under which the maximum entropy formulation aligns with KL-constrained objectives. The resulting objective is given as follows.

$$\max_\pi \mathbb{E}_{x \sim D, y \sim \pi}[r(x, y)] + \alpha D_\mathcal{H}[\pi(y|x)] \tag{10}$$

It is straightforward to show that optimal policy of the equation 10 (proof in Appendix D) is as follows:

$$\pi_r(y|x) = \frac{1}{Z(x)} \exp\left(\frac{1}{\alpha} r(x, y)\right) \tag{11}$$

Following the analytical approach used in DPO's derivation, we can rearrange this optimal policy equation to express the reward function in terms of the policy:

$$r(x, y) = \alpha \log \pi_r(y|x) + \alpha \log Z(x) \tag{12}$$

Now, applying this reparameterization to the Bradley-Terry preference model. For the ground-truth reward $r^*$ and corresponding optimal policy $\pi^*$, the preference probability becomes:

$$p^*(y_1 \succ y_2|x) = \sigma(r^*(x, y_1) - r^*(x, y_2)) \tag{13}$$

Substituting our reparameterization:

$$p^*(y_1 \succ y_2|x) = \sigma\left(\alpha \log \pi^*(y_1|x) + \alpha \log Z(x) - \alpha \log \pi^*(y_2|x) - \alpha \log Z(x)\right) \tag{14}$$

$$= \sigma\left(\alpha \log \pi^*(y_1|x) - \alpha \log \pi^*(y_2|x)\right) \tag{15}$$

Crucially, the partition function $Z(x)$ cancels out, eliminating the need to compute it explicitly. To connect this to SimPO's original formulation, we can decompose the temperature parameter $\alpha$ into two components: $\alpha = \frac{\beta}{|y|}$ where $\beta$ is a scaling factor and $|y|$ provides length normalization.

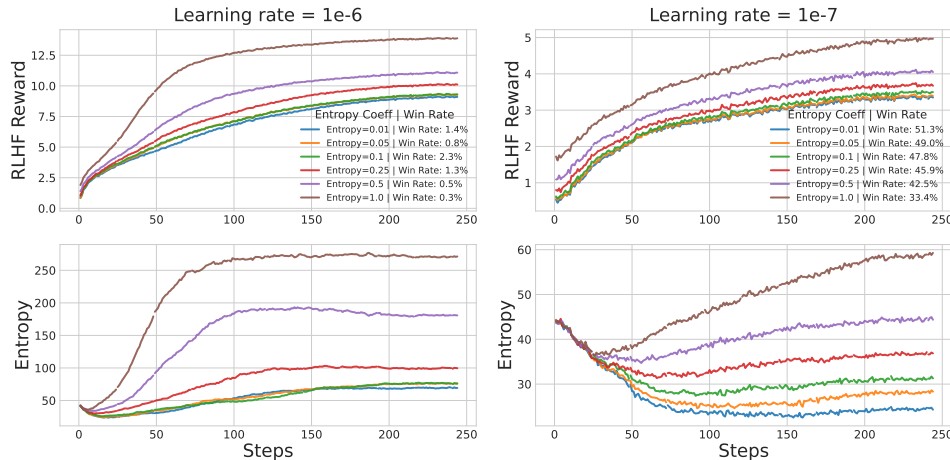

Figure 1: RLHF reward and entropy bonus during training for Pythia 1B with different entropy coefficients at learning rates 1e-6 (left) and 1e-7 (right). Win rates are reported in the legend for each entropy bonus coefficient setting.

Additionally, following $\psi PO$ (Azar et al., 2023), we can augment the objective with a target reward margin $\gamma > 0$ to encourage separation between preferred and dispreferred responses. This leads to the SimPO objective for a parametric policy $\pi_\theta$:

$$\mathcal{L}_{\text{SimPO}}(\pi_\theta) = -\mathbb{E}_{(x,y_w,y_l)\sim\mathcal{D}} \left[ \log \sigma \left( \frac{\beta}{|y_w|} \log \pi_\theta(y_w|x) - \frac{\beta}{|y_l|} \log \pi_\theta(y_l|x) - \gamma \right) \right] \qquad (16)$$

This derivation reveals that SimPO is equivalent to Maximum Entropy RL under the contextual bandit formulation with adaptive temperature and target margin augmentation, making explicit the theoretical connection that underlies SimPO's design. The reference-free nature of SimPO emerges naturally from the Maximum Entropy framework, as no explicit reference policy is required in the entropy-regularized objective. The length normalization can be interpreted as an adaptive temperature parameter that scales inversely with sequence length, while the target margin $\gamma$ encourages better separation between preferences.

**Theoretical Guarantees.** Following the same theoretical framework as DPO, SimPO inherits analogous guarantees regarding representational completeness, equivalence class preservation, and consistency under the Bradley-Terry preference model. The detailed proofs and formal statements of these properties are provided in Appendix D.

## 4 MAXIMUM ENTROPY RLHF

Having established the theoretical connection between SimPO and Maximum Entropy RL, we now turn to the online RLHF setting. Our goal is to evaluate whether Maximum Entropy RL can perform comparably to its KL-constrained counterpart when applied directly to preference optimization.

### 4.1 EXPERIMENTAL SETUP AND METHODOLOGY

In our experiments, we train 1B and 2.8B parameter models from the Pythia suite (Biderman et al., 2023) using RLOO (Ahmadian et al., 2024) on the TL;DR dataset (Stiennon et al., 2022). For optimization, we follow the training recipe outlined in Huang et al. (2024), and implement our experiments using the TRL library (von Werra et al., 2020). We evaluate alignment to human preference using simulated win-rates with GPT-4o-mini (OpenAI et al., 2024) as the proxy evaluator, measured against reference summaries for TL;DR using greedy sampling unless stated otherwise.

Our model and dataset choices are guided by two main considerations. First, our aim is not to train state-of-the-art competitive models, but to study the methodological aspects of the approach.

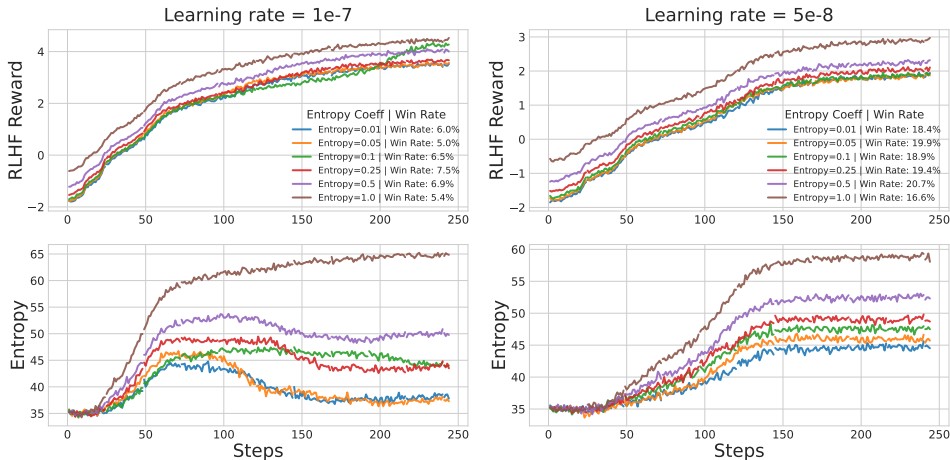

Figure 2: RLHF reward and entropy bonus during training for Pythia 2.8B with different entropy bonus coefficients at learning rates 1e-7 (left) and 5e-8 (right). Win rates are reported in the legend for each entropy bonus coefficient setting.

Second, computational constraints limit us from scaling to larger models. Nonetheless, this setup provides a well-suited testbed for exploring the questions we set out to investigate.

We adopt RLOO as a critic-free alternative to the standard RLHF pipeline, while still optimizing the same underlying objective. In the KL-constrained formulation, the reward is defined as

$$r(x, y) = r_\phi(x, y) - \beta\Big( \log \pi_\theta(y|x) - \log \pi_{\text{ref}}(y|x)\Big), \tag{17}$$

whereas in the length-normalized maximum-entropy variant, it takes the form

$$r(x, y) = r_\phi(x, y) - \frac{\beta}{|y|} \log \pi_\theta(y|x). \tag{18}$$

## 4.2 RESULTS AND ANALYSIS

### 4.2.1 ONLINE MAXIMUM ENTROPY RLHF

**Online Maximum Entropy RLHF with Pythia 1B.** To evaluate the effectiveness of Maximum Entropy RL, inspired by the success of SimPO, we trained the Pythia 1B model across a range of entropy coefficients and two learning rates: $1 \times 10^{-6}$ and $1 \times 10^{-7}$. The entropy coefficients were selected via a simple grid search, while the learning rates were motivated by prior findings, $1 \times 10^{-6}$ being the setting where KL-constrained RLHF performs strongly, and $1 \times 10^{-7}$ following the recommendation from SimPO. Our results, in Figure 1 , reveal that training with $1 \times 10^{-6}$ consistently leads to overoptimization, regardless of the entropy coefficient. This suggests that entropy regularization alone is insufficient to constrain the model, which is ultimately unsurprising.

Nevertheless, we find that lowering the learning rate improves stability and yields reasonably strong results, where well-behaved KL runs achieves win rate of around 50–55%, compared to 30–35% for the SFT baseline. At first glance, this could be interpreted as evidence for the effectiveness of Maximum Entropy RL. However, we observe that even with an entropy coefficient of 0, the model still achieves roughly a 50% win rate, suggesting that the performance gain is not attributable to entropy regularization. Moreover, it is important to note that strong models exhibit decaying and stable entropy bonuses, whereas overoptimized models display increasing entropy bonuses, indicating that entropy actually exacerbates reward hacking rather than mitigating it.

**Online Maximum Entropy RLHF with with Pythia 2.8B.** To further validate our observations, we conducted experiments with Pythia 2.8B, a larger model from the same family. The only change

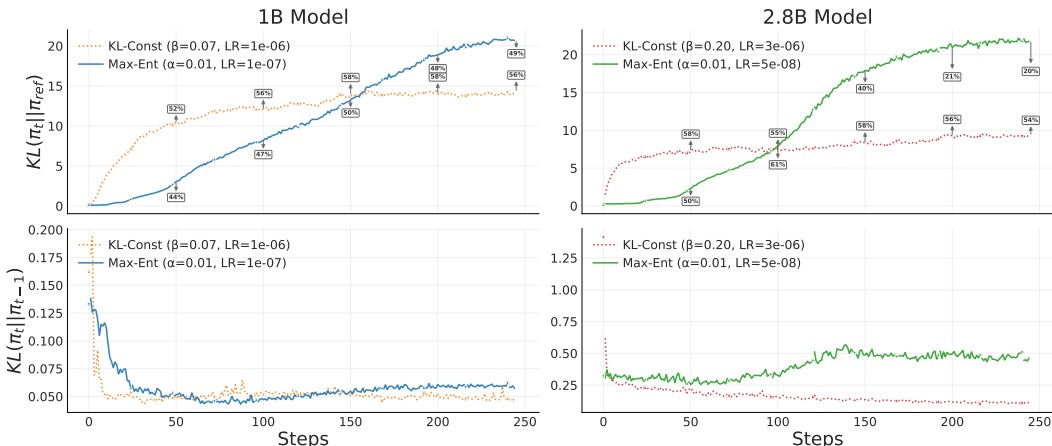

Figure 3: KL divergence metrics and win rates for KL-Constrained and Maximum Entropy regularization methods across training steps. Top row shows KL divergence between current policy and reference policy ($KL(\pi_t||\pi_{\text{ref}})$) for 1B model (left) and 2.8B model (right). Bottom row shows KL divergence between consecutive policy iterations ($KL(\pi_t||\pi_{t-1})$)

in configuration was the learning rate, which we reduced to $1 \times 10^{-7}$ and $5 \times 10^{-8}$ to account for the increased model size; all other hyperparameters were kept as before , and results are given in Figure 2. Surprisingly, none of the trained variants were able to outperform the SFT baseline, as they all exhibited severe overoptimization. Notably, even the very small learning rate of $5 \times 10^{-8}$ led to overoptimization, whereas the KL-constrained approach still achieved strong results with a higher learning rate of $1 \times 10^{-6}$ under the same number of optimization steps. These findings highlight a clear failure case of Maximum Entropy RL as an online RLHF paradigm.

**KL Budget of the Optimization.** It is evident that Maximum Entropy RL is not sufficient to prevent overoptimization, and in fact, even RL without regularization can achieve comparable performance. Nevertheless, we observe overoptimization even at very small learning rates. To investigate this phenomenon, we track the KL divergence between the policy and the reference model during training in the Maximum Entropy setting. In addition, we consider two standard KL-constrained runs: one achieving a strong win rate and another that overoptimizes. The only difference between these two runs is the KL coefficient, which ultimately determines their outcomes. By comparing these cases, we aim to better understand KL behavior under the standard methodology and clarify what constitutes desirable optimization.

A key strength of a well-tuned KL regularizer is that, after some steady improvements, it ensures KL divergence grows only very slowly while the policy remains close to the reference model. This keeps optimization within safe regions. At the same time, KL regularization is highly sensitive, since even small changes in the KL coefficient can cause large shifts that ultimately lead to an overoptimized model. In addition, the appropriate KL weight is not universal and must be carefully tuned for each model, even when trained on the same dataset.

In our Maximum Entropy RL runs with the 1B model, in Figure 5, we find that the policy does not become overoptimized, but its KL grows in a nearly linear fashion, as expected. A similar pattern is observed with the 2.8B model; however, despite ending with almost identical KL values, the 2.8B model still collapses into an overoptimized state. This shows that the optimization budget in reference-free RL methods, whether Maximum Entropy RL or standard RL, is extremely fragile. Even with very small learning rates, models can still undergo significant KL updates that result in an overoptimized outcome.

The reason KL regularization is effective, despite only shaping the reward, is that it maintains a good KL divergence by penalizing out-of-distribution samples and dynamically dampening their effective reward. In contrast, Maximum Entropy RL cannot provide this safeguard because entropy correlates with overoptimization, which amplifies the issue rather than preventing it. Pure RL methods are

even more vulnerable since they directly maximize the proxy reward, which inherently deteriorates once the policy drifts too far from the reference.

**KL Update Magnitudes in Policy Optimization.**    We observe that Pythia 2.8B exhibits high KL updates between consecutive policies even under very low learning rates. One could argue that in Maximum Entropy RL, such high KL updates arise not from the objective itself but from challenges in policy optimization. Since RLOO's policy loss is implemented as PPO (with AC2 being a special case of PPO (Huang et al., 2022)), some may claim the issue is algorithmic, namely PPO's difficulty in keeping ratios bounded (Wang et al., 2020), which ultimately destabilizes KL.

Our results in Figure 3, however, show that this is not purely algorithmic: in KL-constrained runs, PPO successfully maintains stable KL between successive policies, even in overoptimized settings where the KL constraint is relaxed. By contrast, in Maximum Entropy runs we consistently observe increasing KL drift, even in "good" runs, and this effect grows stronger during training despite using very low learning rates compared to KL-constrained runs. A plausible explanation is that unregularized reward optimization produces sharper gradients, which push the policy to change more aggressively. This is somewhat surprising in RLOO, since PPO's average clipping ratio is quite low (unlike in standard RL), and the initial policy is already strong. Yet, in Maximum Entropy settings, we find a quadratic clipping behavior, suggesting that optimization drifts toward reward hacking regions that ignore regularization and focus solely on maximizing reward. To counteract this, we attempted to enforce stricter updates by reducing the PPO clipping parameter $\epsilon$ from the standard 0.2 down to as small as $10^{-4}$ while using a learning rate of $5 \times 10^{-8}$ in Pythia 2.8B experiments. However, this adjustment failed to induce greater pessimism: models still overoptimized, indicating that the problem is not resolved by clipping alone and may indeed be algorithmic.

Overall, our findings suggest that high KL is both an objective-driven and algorithmic phenomenon. KL-constrained runs remain stable (even when overoptimized), while Maximum Entropy runs show persistent KL escalation despite tighter clip ranges. This highlights that the optimization trajectory is strongly shaped by the choice of objective, even when using identical policy optimization techniques.

### 4.2.2    Minimum Entropy RL

Motivated by the link between maximum entropy and overoptimization and recent work showing entropy minimization can serve as an effective reward signal for LLM reasoning (Agarwal et al., 2025), we adopt an unconventional strategy: minimizing entropy to discourage excessively high-entropy which we expect to prevent overoptimization.

Our experiments reveal that Minimum Entropy RL prevents overoptimization and achieves competitive performance with Pythia-1B even at the same learning rate used by KL-constrained RL, under which Maximum Entropy collapses. Yet, with Pythia-2.8B, entropy minimization proves unstable: it is either too conservative, stalling learning, or too loose, leading to overoptimization. While entropy minimization succeeds as a standalone reward for reasoning, combining it with preference-based rewards appears to create optimization instabilities. Reducing the learning rate might offer some improvement, but Minimum Entropy is not a one-to-one substitute for KL, which remains more dynamic and adaptive. Lastly, this underscores that reference-free methods break down once they move outside a healthy KL budget, limiting their reliability.

### 4.3    Offline Maximum Entropy RLHF (SimPO)

Even though Maximum Entropy fails to provide sufficient regularization to prevent overfitting, its closed-form solution, SimPO, proves to be both effective and performant. This effectiveness cannot be solely attributed to the use of a low learning rate, since one of the configurations for Llama 3 (Grattafiori et al., 2024) employs an even higher learning rate than DPO. Nevertheless, maintaining a low learning rate remains critical for controlling the KL, which is a crucial and universal requirement across all alignment algorithms (Gao et al., 2022; Rafailov et al., 2024a).

One might argue that, since all samples are in-distribution, there is no need for an explicit OOD regularizer. However, as noted by Azar et al. (2023); Rafailov et al. (2024a), the reward model effectively drifts out-of-distribution during optimization, which leads to sub-epoch overoptimization. This highlights that the form of the reward model is critical, and that the mere presence of a reference

model is insufficient. To mitigate this, Huang et al. (2025) propose replacing KL with $\chi^2$ regularization, thereby injecting pessimism directly into the reward model. They report that this approach maintains performance across multiple epochs, whereas DPO collapses after just one. However, we were unable to reproduce these results, leaving open the question of whether $\chi^2$ regularization truly implements pessimism. This makes the performance of SimPO particularly intriguing, not because it achieves pessimism, but because it demonstrates strong results even without relying on a reference model.

In DPO, the pairwise reward can be written as $r(y_w|x) - r(y_l|x) = \beta \left( \log \frac{\pi(y_w|x)}{\pi(y_l|x)} - \log \frac{\pi_{\text{ref}}(y_w|x)}{\pi_{\text{ref}}(y_l|x)} \right)$, where the second term reflects the contribution of the reference model. Because both $y_w$ and $y_l$ are sampled from the reference distribution, we expect this term to be negative but small, effectively acting as an *adaptive regularizer*. This parallels the role of a margin in SimPO, with the key distinction that SimPO uses a fixed margin rather than a reference-based one (Ahrabian et al., 2025).

This perspective suggests that offline methods might potentially reduce reliance on reference models by introducing target margins that could serve a similar function to reference contributions. To explore this possibility, we visualize the reference log probability margins $\log \left( \frac{\pi_{\text{ref}}(y_w|x)}{\pi_{\text{ref}}(y_l|x)} \right)$ during DPO training with Pythia 1B, in Figure 4. Our observations suggest that these margins tend to fall within a relatively narrow range, which appears consistent with the fixed margins used in reference free methods like SimPO.

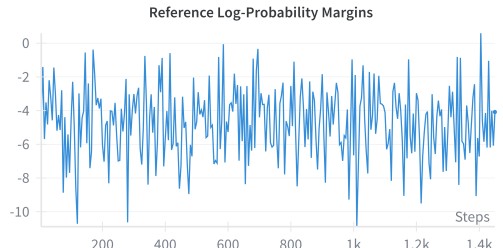

Figure 4: Batch average of $\log\left(\frac{\pi_{\text{ref}}(y_w|x)}{\pi_{\text{ref}}(y_l|x)}\right)$ during DPO training.

Some caveats are worth noting. High learning rates combined with large margins can drive aggressive optimization that maximizes separation, potentially leading to the reward overoptimization behaviors highlighted by Rafailov et al. (2024a). We observe extreme likelihood decreases, suggesting that the model places greater weight on out of distribution samples. Reference models may provide adaptive margins that guide optimization, with minimal contributions reducing overoptimization and larger margins focusing on harder examples. Yet, fixed margins risk forcing overoptimization under aggressive updates. These findings indicate that reference models are neither necessary nor sufficient to prevent overoptimization, as reward hacking in DAAs still emerges from overfitting to the reward objective, with cancellation effects limiting protection. We discuss these dynamics in more detail in the Appendix B.

## 5 CONCLUSION

This work establishes a theoretical foundation for SimPO by connecting it to Maximum Entropy Reinforcement Learning with adaptive temperature scaling, while revealing a striking asymmetry between offline and online performance. Although SimPO excels in offline preference optimization, our empirical investigation shows that online Maximum Entropy RL suffers from instability and overoptimization, with entropy regularization paradoxically correlating with rather than preventing reward hacking. These findings highlight that reference-free approaches, while appealing for their simplicity, may face fundamental limitations in online training scenarios, and suggest that SimPO's success stems from implicit stabilizing factors such as dataset constraints and target margins that approximate the regularization benefits of reference models.

## ETHICS STATEMENT

This work focuses on the theoretical and empirical analysis of reinforcement learning objectives for aligning large language models. All experiments were conducted on publicly available preference datasets, and no personally identifiable or sensitive information was used. Our results are intended to improve the understanding of alignment methods and do not involve deployment of models in real-

world settings. Nevertheless, as with all research on large language models, advances in alignment can have dual-use implications: while they may contribute to safer and more reliable AI systems, they could also lower barriers to developing more capable models that might be misused. We encourage responsible use and further investigation into the societal impacts of alignment research.

## REPRODUCIBIBLITY STATEMENT

Our experiments are based on publicly available models (Pythia (Biderman et al., 2023)) and the TRL library (von Werra et al., 2020), with only minimal modifications. Because we rely primarily on standard, open-source components, our results are fully reproducible and can be replicated by other researchers.

## THE USE OF LARGE LANGUAGE MODELS

All text was initially drafted by the authors, after which Large Language Models were employed to refine phrasing and enhance clarity of expression.

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

## A ADDITIONAL RELATED WORK

**Reference-free Alignment.** While early methods like RRHF (Yuan et al., 2023) and RAFT (Dong et al., 2023) still relied on external reward models for ranking, they revealed that complex RL dynamics were unnecessary. SLiC-HF (Zhao et al., 2023) showed that sequence likelihood calibration could directly incorporate human feedback without explicit reward modeling. ORPO (Hong et al., 2024) made the key insight that odds ratios could replace probability ratios, enabling monolithic training without reference model drift. CPO (Xu et al., 2024a) and SimPO (Meng et al., 2024) both recognized that sequence probabilities themselves encode preference signals. SimPO can be seen as CPO's length-normalized variant with zero behavior cloning, but this seemingly minor change eliminates the need for hyperparameter tuning of the BC coefficient. The Cringe Loss (Xu et al., 2024b) explored iterative self-improvement through token-level soft margins rather than sequence-level optimization. The proliferation of SimPO variants (AlphaPO's (Gupta et al., 2025) reward shaping, $\gamma$PO's adaptive margins (Sun et al., 2025), AMoPO's (Liu et al., 2025) multi-objective extension, ConfPO's (Yoon et al., 2025) token-level refinement) demonstrates the flexibility of SimPO's reward formulation while addressing specific optimization challenges.

**Overoptimization in Preference Learning** Reward hacking (Skalse et al., 2025) is a long-standing problem in reinforcement learning (Sutton & Barto, 2018) where policies achieve high rewards but fail to meet the actual objective (Amodei et al., 2016; Hadfield-Menell et al., 2020; Pan et al., 2022). In language model alignment, this manifests as models learning to generate outputs that score highly on proxy metrics while being of poor actual quality. This overoptimization phenomenon was first systematically studied in traditional RLHF (Christiano et al., 2023; Stiennon et al., 2022; Gao et al., 2022; Ouyang et al., 2022), where optimizing imperfect proxy reward models leads to qualitatively worse outputs, including overly wordy responses and hallucinated information.

Direct alignment algorithms like DPO (Rafailov et al., 2024c) were designed to bypass RL training by parameterizing rewards directly in terms of the policy, but they introduce their own form of overoptimization. Azar et al. (2023) show that DPO's unbounded log-odds transformation leads to severely overfitted implicit rewards, losing the regularization benefits of standard RLHF's explicit reward modeling. They propose IPO using bounded $\Psi$ functions to address this issue. However, Rafailov et al. (2024a) demonstrate that even IPO, despite its theoretical guarantees against overoptimization, still exhibits similar degradation patterns to DPO and RLHF at higher KL budgets and across different model scales, suggesting that overoptimization in direct alignment algorithms may be a more fundamental issue than initially anticipated. More recently, Huang et al. (2025) propose $\chi^2$-Preference Optimization ($\chi$PO), which replaces DPO's logarithmic link function with $\chi^2$-divergence regularization to implement pessimism under uncertainty, providing theoretical guarantees against overoptimization based on single-policy concentrability.

## B MARGINS AND OVEROPTIMIZATION

It has been shown that methods such as SimPO can achieve performance comparable to DPO even with a target margin of $\gamma = 0$, as demonstrated in the original SimPO paper. This suggests that offline methods do not necessarily require reference models when operating within the safe KL region, and that introducing margins generally improves performance across benchmarks. This effect arises from both model capabilities and dataset coverage: larger models are less prone to common overfitting behaviors and can extract more meaningful signals during optimization, rather than engaging in reward hacking, a phenomenon observed in both online and offline preference optimization (Gao et al., 2022; Rafailov et al., 2024c). Consequently, the influence of the reference model is minimal and can often be neglected. However, this behavior is contingent on the task being sufficiently challenging and the model being strong enough to avoid overoptimization. To validate this observation, we train Pythia-1B on TL;DR using SimPO across different margin values ($\gamma$) and learning rates, in a setting where the model is relatively weaker and the task is easier compared to standard chat datasets such as UltraFeedback (Cui et al., 2024) used in SimPO.

We first consider a learning rate of $1 \times 10^{-6}$, which is known to be effective for DPO, DPO metrics in Figure 9. In this setting, all SimPO models exhibit overoptimization regardless of the $\gamma$ hyperparameter, SimPO metrics in Figure 8 . Although reward definitions differ and direct comparison of losses or other training metrics is challenging, log-probabilities of samples remain comparable.

We observe the characteristic extreme likelihood decreases, which correlate with overoptimization; this pattern is present in DAAs and, as we show, also occurs in online methods. Increasing the margin exacerbates this issue, as optimization aggressively seeks high separation, naturally resulting in overoptimization.

Reference-free methods like SimPO are particularly susceptible because they lack prior knowledge about sample difficulty, treating all samples equally. Some samples are inherently harder and should receive more attention, a behavior that could be partially captured by negative reference contributions in pairwise preferences. When a hardcoded margin pushes the model to satisfy strict separation objectives, it can amplify pathological behaviors during training.

However, when using a relatively low learning rate that allows for gradual updates, SimPO performs significantly better, metrics in Figure 7 and win rates in Figure 6 . In this regime, it emerges as a strong preference optimization method: an appropriate margin encourages the model to learn and optimize meaningful signals. Therefore, reference-free models require extra safeguards against overoptimization. Controlling the learning rate can act as an anchor, keeping updates within meaningful distributional shifts, although these models can still experience the overoptimization patterns observed in DAAs.

## C    EXTRA FIGURES

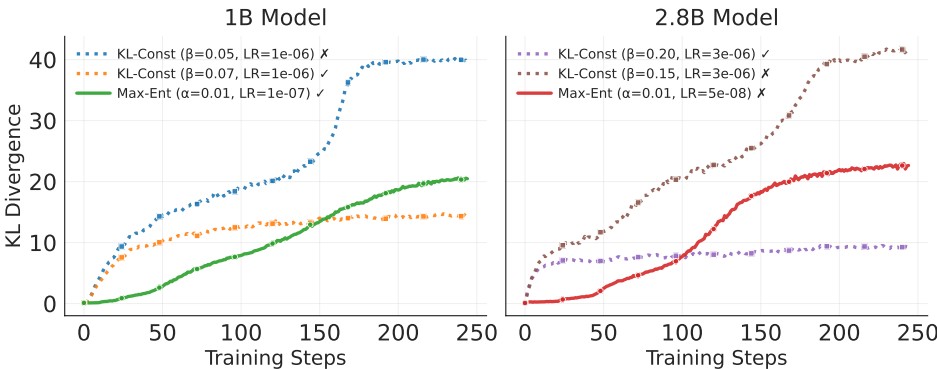

Figure 5: KL divergence evolution during training for 1B and 2.8B parameter models using different regularization methods. The left panel shows results for the 1B model and the right panel shows results for the 2.8B model. Each panel compares KL-Constrained and Maximum-Entropy approaches. Checkmarks (✓) indicate high win rate runs and crosses (×) indicate overoptimized runs.

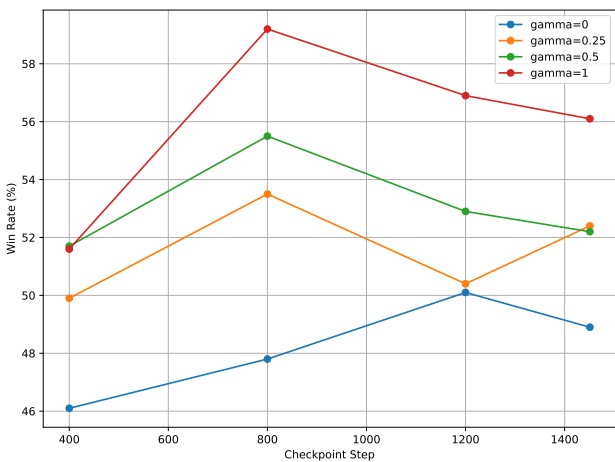

Figure 6: Win rate progression across training checkpoints for different values of the gamma hyper-parameter. Results are for the Pythia-1B model trained with a learning rate of $2 \times 10^{-7}$.

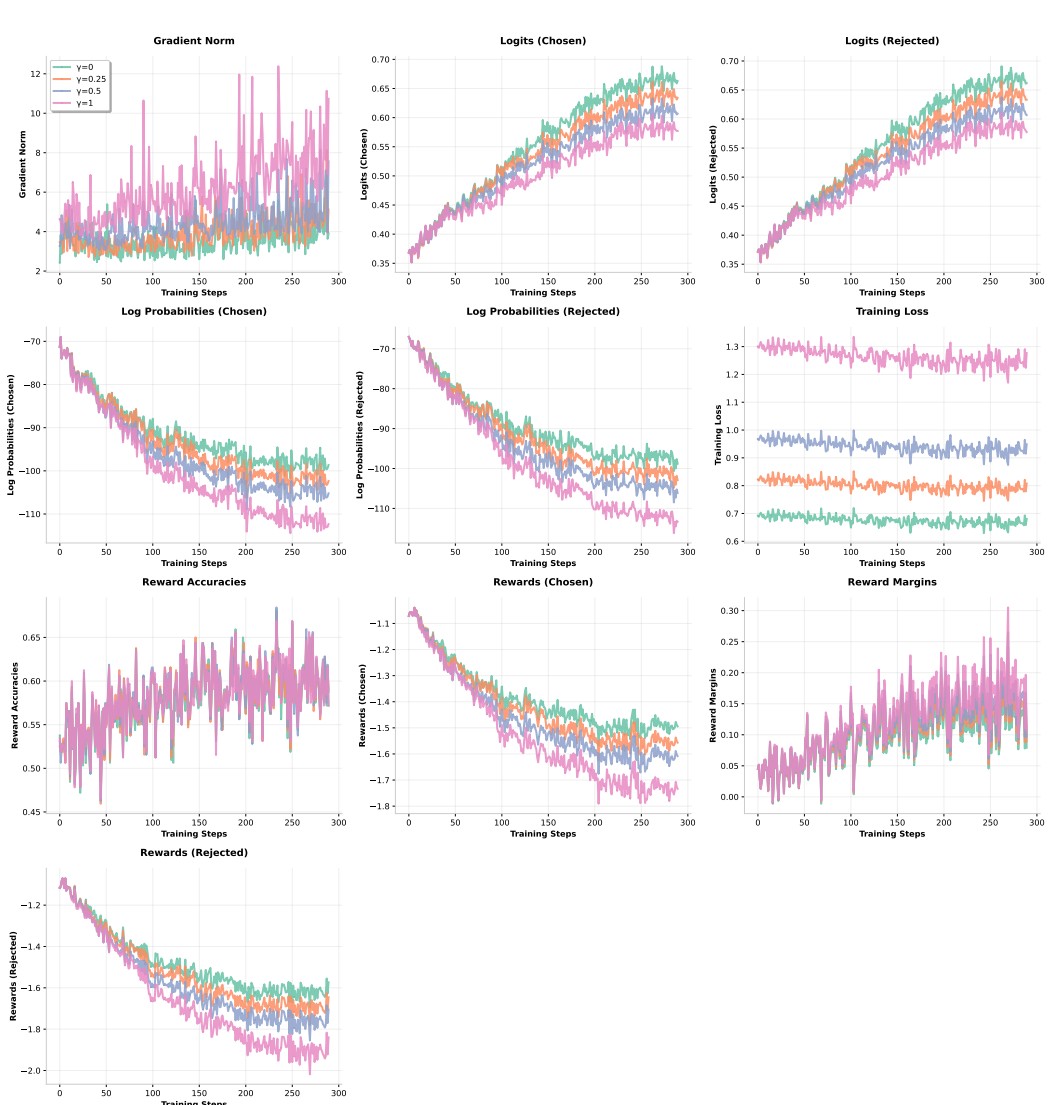

Figure 7: SimPO training metrics across different gamma values. Comparison of key training dynamics including loss, gradients, logits, and reward metrics for $\gamma \in \{0, 0.25, 0.5, 1.0\}$ using Pythia-1B with learning rate $2 \times 10^{-7}$.

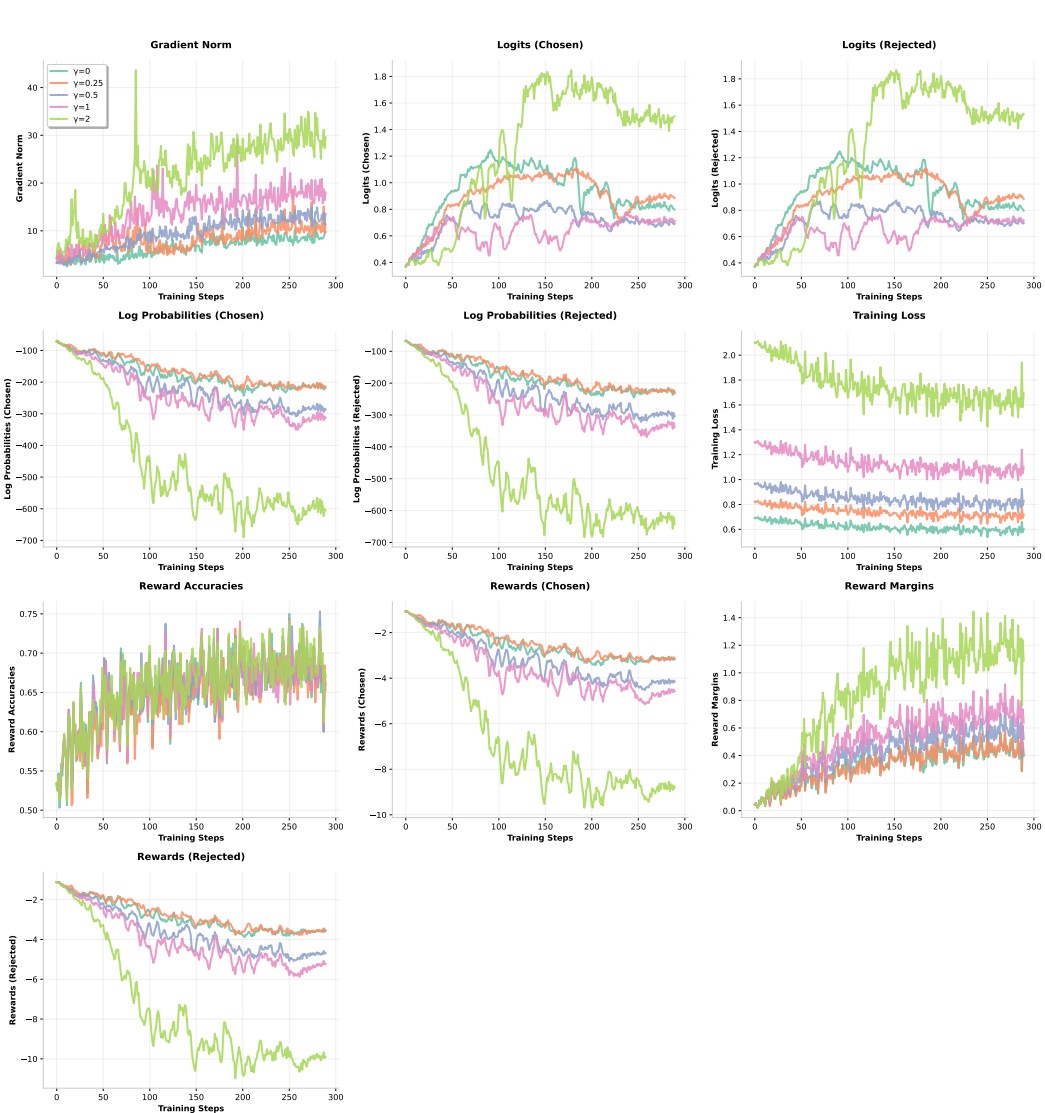

Figure 8: SimPO training metrics across different gamma values. Comparison of key training dynamics including loss, gradients, logits, and reward metrics for $\gamma \in \{0, 0.25, 0.5, 1.0, 2.0\}$ using Pythia-1B with learning rate $1 \times 10^{-6}$.

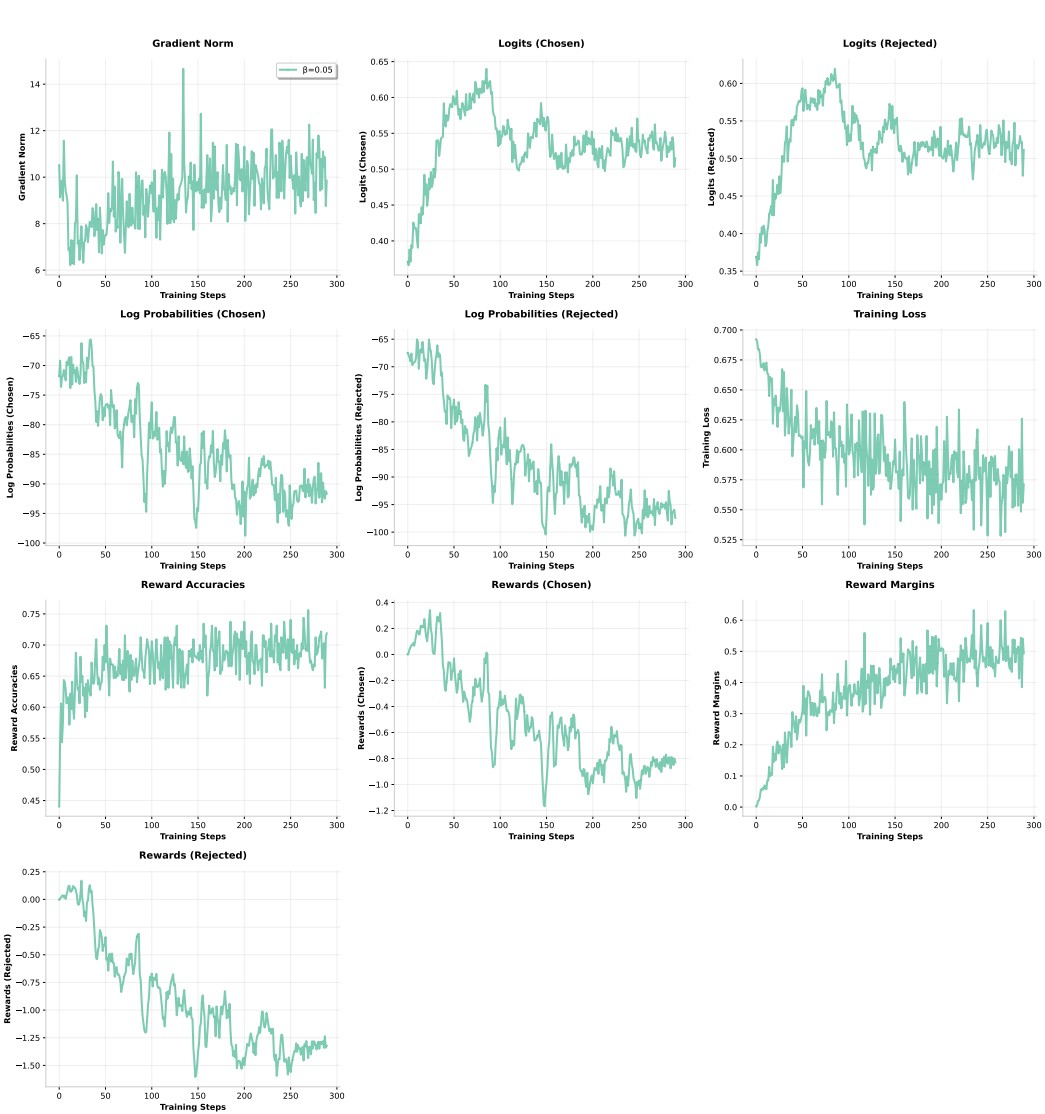

Figure 9: DPO training metrics with $\beta = 0.05$. Comparison of key training dynamics including loss, gradients, logits, and reward metrics, using Pythia-1B with learning rate $1 \times 10^{-6}$.

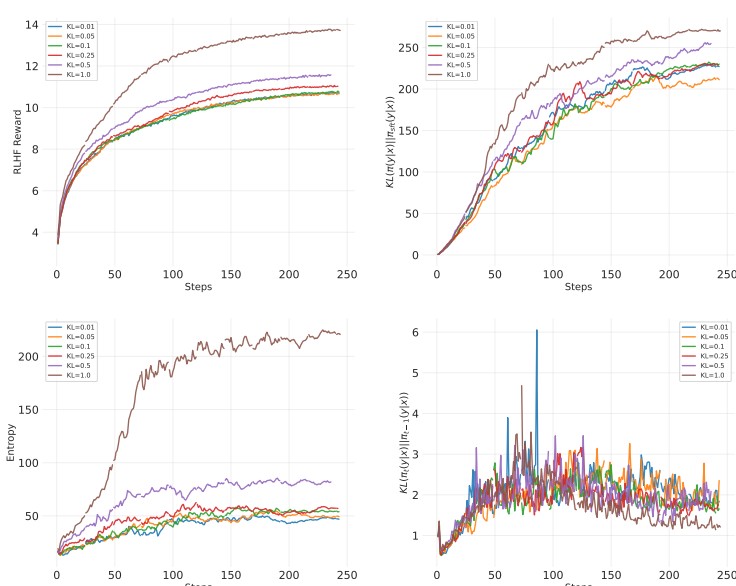

Figure 10: Reward dynamics and KL divergence metrics for entropy-regularized RL training across different entropy coefficients. Top-left panel shows reward progression (RLHF reward) over training steps for various entropy values. Top-right panel shows KL divergence between the current policy and the SFT reference policy ($KL(\pi_t||\pi_{\text{SFT}})$). Bottom-left panel tracks entropy reward across training steps. Bottom-right panel displays KL divergence between consecutive policy updates ($KL(\pi_t||\pi_{t-1})$). All plots are based on the Pythia-6.9B model trained with the learning rate of $1 \times 10^{-6}$.

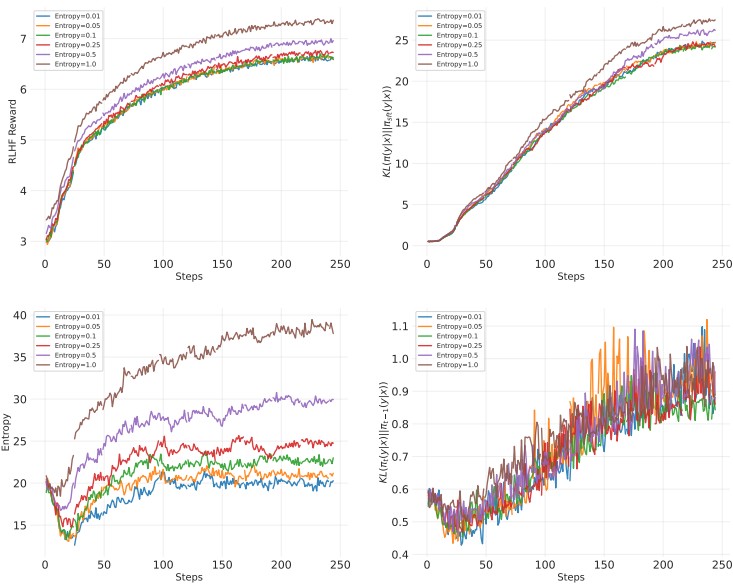

Figure 11: Reward dynamics and KL divergence metrics for entropy-regularized RL training across different entropy coefficients. Top-left panel shows reward progression (RLHF reward) over training steps for various entropy values. Top-right panel shows KL divergence between the current policy and the SFT reference policy ($KL(\pi_t||\pi_{\text{SFT}})$). Bottom-left panel tracks entropy reward across training steps. Bottom-right panel displays KL divergence between consecutive policy updates ($KL(\pi_t||\pi_{t-1})$). All plots are based on the Pythia-6.9B model trained with the learning rate of $1 \times 10^{-7}$.

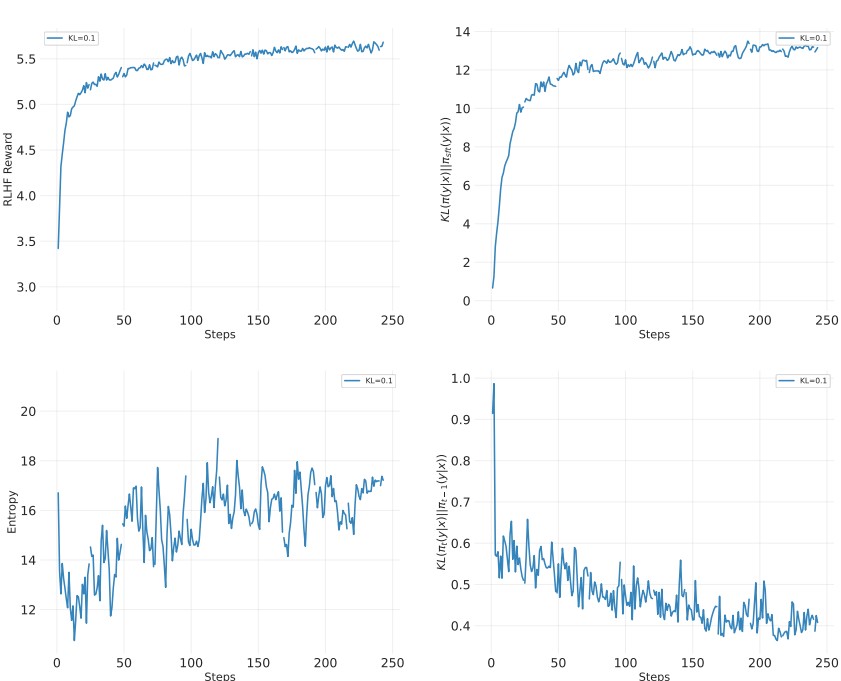

Figure 12: Training metrics for KL-constrained RL on the Pythia-6.9B model. Top panel shows the KL divergence between the policy and reference SFT policy ($KL(\pi_t||\pi_{\text{SFT}})$) over training steps. Middle panel displays the reward trajectory (RLHF reward). Bottom panel shows the KL divergence between consecutive policy updates ($KL(\pi_t||\pi_{t-1})$). All results correspond to a single training run with a fixed KL constraint.

## D  MATHEMATICAL DERIVATIONS FOR MAXIMUM ENTROPY RL

### D.1  DERIVING THE OPTIMUM OF THE ENTROPY-REGULARIZED REWARD MAXIMIZATION OBJECTIVE

In this appendix, we will derive the optimal policy for Maximum Entropy RL. Analogously to the KL-constrained case (Rafailov et al., 2024c), we optimize the following objective:

$$\max_{\pi} \mathbb{E}_{x \sim \mathcal{D}, y \sim \pi} \big[ r(x, y) \big] + \alpha \mathcal{H} \big[ \pi(y|x) \big] \tag{19}$$

under any reward function $r(x, y)$ and a general non-parametric policy class, where $\mathcal{H}[\pi(y|x)] = -\mathbb{E}_{y \sim \pi(y|x)}[\log \pi(y|x)]$ is the entropy of the policy. We now have:

$$\max_{\pi} \mathbb{E}_{x \sim \mathcal{D}, y \sim \pi} \big[ r(x, y) \big] + \alpha \mathcal{H} \big[ \pi(y|x) \big]$$

$$= \max_{\pi} \mathbb{E}_{x \sim \mathcal{D}} \mathbb{E}_{y \sim \pi(y|x)} \big[ r(x, y) - \alpha \log \pi(y|x) \big]$$

$$= \min_{\pi} \mathbb{E}_{x \sim \mathcal{D}} \mathbb{E}_{y \sim \pi(y|x)} \left[ \log \pi(y|x) - \frac{1}{\alpha} r(x, y) \right]$$

$$= \min_{\pi} \mathbb{E}_{x \sim \mathcal{D}} \mathbb{E}_{y \sim \pi(y|x)} \left[ \log \frac{\pi(y|x)}{\frac{1}{Z(x)} \exp\left(\frac{1}{\alpha} r(x, y)\right)} - \log Z(x) \right] \tag{20}$$

where we have partition function:

$$Z(x) = \sum_{y} \exp\left(\frac{1}{\alpha} r(x, y)\right).$$

Note that the partition function is a function of only $x$ and the reward function $r$, but does not depend on the policy $\pi$. We can now define

$$\pi^*(y|x) = \frac{1}{Z(x)} \exp\left(\frac{1}{\alpha} r(x, y)\right),$$

which is a valid probability distribution as $\pi^*(y|x) \geq 0$ for all $y$ and $\sum_y \pi^*(y|x) = 1$. Since $Z(x)$ is not a function of $y$, we can then re-organize the final objective in Eq 20 as:

$$\min_{\pi} \mathbb{E}_{x \sim \mathcal{D}} \left[ \mathbb{E}_{y \sim \pi(y|x)} \left[ \log \frac{\pi(y|x)}{\pi^*(y|x)} \right] - \log Z(x) \right] = \tag{21}$$

$$\min_{\pi} \mathbb{E}_{x \sim \mathcal{D}} \big[ \mathbb{D}_{\text{KL}}(\pi(y|x) \,||\, \pi^*(y|x)) - \log Z(x) \big] \tag{22}$$

Since $Z(x)$ is independent of $\pi$, the minimum is attained by the policy that minimizes the first KL term. By Gibbs' inequality, the KL divergence reaches its minimum value of zero if and only if the two distributions are identical. Therefore, this yields the optimal solution. :

$$\pi(y|x) = \pi^*(y|x) = \frac{1}{Z(x)} \exp\left(\frac{1}{\alpha} r(x, y)\right) \tag{23}$$

for all $x \in \mathcal{D}$. This completes the derivation.

### D.2  DERIVING THE SIMPO OBJECTIVE UNDER THE BRADLEY-TERRY MODEL

It is straightforward to derive the SimPO objective under the Bradley-Terry preference model as we have

$$p^*(y_1 \succ y_2 | x) = \frac{\exp\left(r^*(x, y_1)\right)}{\exp\left(r^*(x, y_1)\right) + \exp\left(r^*(x, y_2)\right)} \tag{24}$$

We can express the (unavailable) ground-truth reward through its corresponding optimal policy:

$$r^*(x, y) = \alpha \log \pi^*(y|x) + \alpha \log Z(x) \tag{25}$$

Substituting Eq. 25 into Eq. 24 we obtain:

$$p^*(y_1 \succ y_2|x) = \frac{\exp\left(\alpha \log \pi^*(y_1|x) + \alpha \log Z(x)\right)}{\exp\left(\alpha \log \pi^*(y_1|x) + \alpha \log Z(x)\right) + \exp\left(\alpha \log \pi^*(y_2|x) + \alpha \log Z(x)\right)}$$

$$= \frac{1}{1 + \exp\left(\alpha \log \pi^*(y_2|x) - \alpha \log \pi^*(y_1|x)\right)}$$

$$= \sigma\left(\alpha \log \pi^*(y_1|x) - \alpha \log \pi^*(y_2|x)\right).$$

The last line is the per-instance loss for SimPO, without target margin $\gamma$ and length normalization.

### D.3 DERIVING THE SIMPO OBJECTIVE UNDER THE PLACKETT-LUCE MODEL

The Plackett-Luce model (Plackett, 1975) extends the Bradley-Terry model from pairwise comparisons to full rankings. As in the Bradley-Terry framework, the probability of selecting an option is assumed to be proportional to the value of an underlying latent reward function. In our setting, given a prompt $x$ and a collection of $K$ candidate answers $y_1, \ldots, y_K$, the user produces a permutation $\tau : [K] \to [K]$ that represents their ranking of the answers. Under the Plackett-Luce model, the probability of such a ranking is defined as follows:

$$p^*(\tau|y_1, \ldots, y_K, x) = \prod_{k=1}^{K} \frac{\exp(r^*(x, y_{\tau(k)}))}{\sum_{j=k}^{K} \exp(r^*(x, y_{\tau(j)}))} \tag{26}$$

Observe that when $K = 2$, Equation 26 simplifies to the Bradley-Terry model. For the general Plackett-Luce model, however, we can still leverage the reward parameterization by substituting the reward function expressed in terms of its optimal policy. As in Appendix D.2, the normalization constant $Z(x)$ cancels out, leaving us with:

$$p^*(\tau|y_1, \ldots, y_K, x) = \prod_{k=1}^{K} \frac{\exp\left(\alpha \log \pi^*(y_{\tau(k)}|x)\right)}{\sum_{j=k}^{K} \exp\left(\alpha \log \pi^*(y_{\tau(j)}|x)\right)} \tag{27}$$

Similarly to the approach for standard DPO, if we have access to a dataset $\mathcal{D} = \{\tau^{(i)}, y_1^{(i)}, \ldots, y_K^{(i)}, x^{(i)}\}_{i=1}^{N}$ of prompts and user-specified rankings, we can use a parameterized model and optimize this objective with maximum-likelihood.:

$$\mathcal{L}_{\text{SimPO}}(\pi_\theta) = -\mathbb{E}_{\tau, y_1, \ldots, y_K, x \sim \mathcal{D}} \left[ \log \prod_{k=1}^{K} \frac{\exp\left(\alpha \log \pi_\theta(y_{\tau(k)}|x)\right)}{\sum_{j=k}^{K} \exp\left(\alpha \log \pi_\theta(y_{\tau(j)}|x)\right)} \right] \tag{28}$$

### D.4 DERIVING THE GRADIENT OF THE SIMPO OBJECTIVE

In this section we derive the gradient of the SimPO objective:

$$\nabla_\theta \mathcal{L}_{\text{SimPO}}(\pi_\theta) = -\nabla_\theta \mathbb{E}_{(x, y_w, y_l) \sim \mathcal{D}} \left[ \log \sigma\left(\alpha \log \pi_\theta(y_w|x) - \alpha \log \pi_\theta(y_l|x)\right) \right] \tag{29}$$

We can rewrite the RHS of Equation 29 as

$$\nabla_\theta \mathcal{L}_{\text{SimPO}}(\pi_\theta) = -\mathbb{E}_{(x, y_w, y_l) \sim \mathcal{D}} \left[ \frac{\sigma'(u)}{\sigma(u)} \nabla_\theta(u) \right], \tag{30}$$

where $u = \alpha \log \pi_\theta(y_w|x) - \alpha \log \pi_\theta(y_l|x)$.

Using the properties of sigmoid function $\sigma'(x) = \sigma(x)(1 - \sigma(x))$ and $\sigma(-x) = 1 - \sigma(x)$, we obtain the final gradient

$$\nabla_\theta \mathcal{L}_{\text{SimPO}}(\pi_\theta) =$$

$$- \mathbb{E}_{(x, y_w, y_l) \sim \mathcal{D}} \left[ \alpha \sigma\left(\alpha \log \pi_\theta(y_l|x) - \alpha \log \pi_\theta(y_w|x)\right) \left[ \nabla_\theta \log \pi(y_w \mid x) - \nabla_\theta \log \pi(y_l \mid x) \right] \right],$$

After using the reward substitution of $\hat{r}_\theta(x, y) = \alpha \log \pi_\theta(y \mid x)$ we obtain the final form of the gradient.

### D.5 PROOF OF LEMMA 1 AND 2 FROM DPO FOR MAXIMUM ENTROPY RL

In this section, we will prove the two lemmas from DPO for Maximum Entropy RL.

**Lemma 1** (Lemma 1). *Under the Plackett-Luce preference framework, and in particular the Bradley-Terry framework, two reward functions from the same equivalence class induce the same preference distribution.*

*Proof.* We say that two reward functions $r(x, y)$ and $r'(x, y)$ are from the same equivalence class if $r'(x, y) = r(x, y) + f(x)$ for some function $f$. We consider the general Plackett-Luce (with the Bradley-Terry model a special case for $K = 2$) and denote the probability distribution over rankings induced by a particular reward function $r(x, y)$ as $p_r$. For any prompt $x$, answers $y_1, \ldots, y_K$ and ranking $\tau$ we have:

$$
\begin{aligned}
p_{r'}(\tau|y_1, \ldots, y_K, x) &= \prod_{k=1}^{K} \frac{\exp(r'(x, y_{\tau(k)}))}{\sum_{j=k}^{K} \exp(r'(x, y_{\tau(j)}))} \\
&= \prod_{k=1}^{K} \frac{\exp(r(x, y_{\tau(k)}) + f(x))}{\sum_{j=k}^{K} \exp(r(x, y_{\tau(j)}) + f(x))} \\
&= \prod_{k=1}^{K} \frac{\exp(f(x)) \exp(r(x, y_{\tau(k)}))}{\exp(f(x)) \sum_{j=k}^{K} \exp(r(x, y_{\tau(j)}))} \\
&= \prod_{k=1}^{K} \frac{\exp(r(x, y_{\tau(k)}))}{\sum_{j=k}^{K} \exp(r(x, y_{\tau(j)}))} \\
&= p_r(\tau|y_1, \ldots, y_K, x),
\end{aligned}
$$

which completes the proof. $\square$

**Lemma 2** (Lemma 2). *Two reward functions from the same equivalence class induce the same optimal policy under the entropy-regularized RL problem.*

*Proof.* Let us consider two reward functions from the same class, such that $r'(x, y) = r(x, y) + f(x)$ and, let us denote as $\pi_r$ and $\pi_{r'}$ the corresponding optimal policies. For all $x, y$ we have

$$
\begin{aligned}
\pi_{r'}(y|x) &= \frac{1}{\sum_y \exp\left(\frac{1}{\alpha} r'(x, y)\right)} \exp\left(\frac{1}{\alpha} r'(x, y)\right) \\
&= \frac{1}{\sum_y \exp\left(\frac{1}{\alpha}(r(x, y) + f(x))\right)} \exp\left(\frac{1}{\alpha}(r(x, y) + f(x))\right) \\
&= \frac{1}{\exp\left(\frac{1}{\alpha} f(x)\right) \sum_y \exp\left(\frac{1}{\alpha} r(x, y)\right)} \exp\left(\frac{1}{\alpha} r(x, y)\right) \exp\left(\frac{1}{\alpha} f(x)\right) \\
&= \frac{1}{\sum_y \exp\left(\frac{1}{\alpha} r(x, y)\right)} \exp\left(\frac{1}{\alpha} r(x, y)\right) \\
&= \pi_r(y|x),
\end{aligned}
$$

which completes the proof. $\square$

### D.6 PROOF OF THEOREM 1 FROM DPO FOR MAXIMUM ENTROPY RL

In this section, we will eloborate on the results of the main theorem from DPO for Maximum Entropy RL.

**Theorem 1** (Maximum Entropy Version). *Assume we have a parameter $\alpha > 0$. All reward equivalence classes, as defined in the previous section, can be represented with the reparameterization $r(x, y) = \alpha \log \pi(y|x)$ for some model $\pi(y|x)$.*

*Proof.* Consider any reward function $r(x, y)$, which induces an optimal model $\pi_r(y|x)$ under the entropy-regularized RL problem, with solution given by the optimal policy derivation. We have:

$$r(x, y) = \alpha \log \pi_r(y|x) + \alpha \log Z(x)$$

where $Z(x) = \sum_y \exp\left(\frac{1}{\alpha} r(x, y)\right)$ (notice that $Z(x)$ also depends on the reward function $r$). Using the operator $r'(x, y) = f(r, \alpha)(x, y) = r(x, y) - \alpha \log Z(x)$, we see that this new reward function is within the equivalence class of $r$ and, we have:

$$r'(x, y) = \alpha \log \pi_r(y|x)$$

which completes the proof. $\qquad\square$

We can further expand on these results. We can see that if $r$ and $r'$ are two reward functions in the same class, then

$$f(r, \alpha)(x, y) = \alpha \log \pi_r(y|x) = \alpha \log \pi_{r'}(y|x) = f(r', \alpha)(x, y)$$

where the second equality follows from Lemma 2. We have proven that the operator $f$ maps all reward functions from a particular equivalence class to the same reward function. Next, we show that for every equivalence class of reward functions, the reward function that has the reparameterization outlined in the main theorem is unique.

**Proposition 1.** *Assume we have a parameter $\alpha > 0$. Then every equivalence class of reward functions has a unique reward function $r(x, y)$, which can be reparameterized as $r(x, y) = \alpha \log \pi(y|x)$ for some model $\pi(y|x)$.*

*Proof.* We will proceed using proof by contradiction. Assume we have two reward functions from the same class, such that $r'(x, y) = r(x, y) + f(x)$. Moreover, assume that $r'(x, y) = \alpha \log \pi'(y|x)$ for some model $\pi'(y|x)$ and $r(x, y) = \alpha \log \pi(y|x)$ for some model $\pi(y|x)$, such that $\pi \neq \pi'$. We then have

$$r'(x, y) = r(x, y) + f(x) = \alpha \log \pi(y|x) + f(x) = \alpha \log \pi(y|x) \exp(\frac{1}{\alpha} f(x)) = \alpha \log \pi'(y|x)$$

for all prompts $x$ and completions $y$. Then we must have $\pi(y|x) \exp(\frac{1}{\alpha} f(x)) = \pi'(y|x)$. Since these are distributions, summing over $y$ on both sides, we obtain that $\exp(\frac{1}{\alpha} f(x)) = 1$ and since $\alpha > 0$, we must have $f(x) = 0$ for all $x$. Therefore $r(x, y) = r'(x, y)$. This completes the proof. $\qquad\square$

We have now shown that every reward class has a unique reward function that can be represented as outlined in the main theorem, which is given by $f(r, \alpha)$ for any reward function in that class.

# E   PYTHIA 6.9B RESULTS

In addition to our results on the 1B and 2.8B models, we also evaluated the 6.9B model, which is the largest model from Huang et al. (2024). Its behavior exhibits a mixture of the patterns we observed in the smaller models. Figure 10 shows the run with a learning rate of $1 \times 10^{-6}$, and Figure 11 shows the run with a learning rate of $1 \times 10^{-7}$.

First, the 6.9B model performs well at a learning rate of $1 \times 10^{-7}$, but fully optimizes at $1 \times 10^{-6}$, where the KL constrained method achieves the best performance while spending a very small KL budget. This differs from the 2.8B model and suggests that models do not necessarily operate under similar effective KL budgets. As a result, Maximum Entropy and other methods without explicit anchoring to a reference policy are prone to overoptimization. We also note that none of the runs at the $1 \times 10^{-7}$ learning rate overoptimized, whereas the $1 \times 10^{-6}$ runs consistently did.

Second, we observe that the model updates more aggressively, similar to the 2.8B model, where consecutive updates grow in magnitude. This indicates that the model undergoes larger parameter shifts and follows a noticeably different optimization trajectory. This behavior is further supported by the KL patterns shown in Figure 12. While Maximum Entropy produces a roughly linear KL increase, we would expect a more sigmoidal shape due to the decaying learning rate schedule, and the KL constrained runs reflect this expected behavior.

