# OpenReview forum: "Failure Modes of Maximum Entropy RLHF"
_ICLR.cc/2026/Conference — ICLR 2026 Conference Withdrawn Submission_

### Official Review · Reviewer_4ozv · 2025-10-24

**Soundness:** 2
**Presentation:** 3
**Contribution:** 2
**Rating:** 4
**Confidence:** 3

**Summary:**

This paper investigates an asymmetry in preference optimization by first providing a novel theoretical foundation for the offline algorithm SimPO, deriving it from a Maximum Entropy (MaxEnt) RL framework. Motivated by this, the authors empirically test a standard online MaxEnt RLHF objective and find that it consistently fails. Unlike stable KL-constrained methods, the online MaxEnt approach suffers from severe overoptimization and reward hacking, with entropy regularization appearing to correlate with—rather than prevent—these instabilities.

**Strengths:**

1. The paper provides a significant theoretical contribution by deriving SimPO from a Maximum Entropy RL framework. This establishes a principled foundation for SimPO, which, unlike DPO, previously lacked a clear theoretical explanation.
2. This paper investigates that MaxEnt RLHF consistently fails due to overoptimization and entropy regularization correlates with reward hacking.

**Weaknesses:**

1. The paper's central claim of equivalence between SimPO and MaxEnt RL appears non-rigorous. The derivation (Eq. 14) relies on the cancellation of the partition function term $\alpha \log Z(x)$. However, this only holds for a constant α. The paper's linkage to SimPO uses a length-normalized temperature $\alpha = \beta / |y|$, which depends on the response y. Since $|y_w|$ and |$y_l|$ are generally not equal, their corresponding α terms differ, meaning the $\alpha \log Z(x)$  terms would not cancel. This is a critical point, as the original SimPO paper showed that length normalization is key to its performance.
2. The empirical claims about online failure, while strong, are based on experiments with Pythia-1B/2.8B models and the TL;DR summarization task. It is unclear if these same failure modes would persist on larger, more capable models (e.g., 7B+) or on more complex alignment tasks.

**Questions:**

1. Regarding the theoretical derivation (Eq. 14): Could you clarify how the $\alpha \log Z(x)$ terms cancel given that the link to SimPO requires a length-normalized temperature $\alpha = \beta / |y|$? Since $|y_w|$ and $|y_l|$ are not generally equal, their corresponding α values would differ. Does the theoretical link to MaxEnt RL still hold if this is corrected?
2. In line 320, the authors mention that "entropy actually exacerbates reward hacking rather than mitigating it." This claim seems counter-intuitive, since reward hacking is often caused by a collapsed distribution. Could the authors further explain this point?

---

> ### Author Response · Authors · 2025-11-18
>
> We thank the reviewer for this careful technical analysis and the opportunity to clarify our work.
>
> On the Partition Function Cancellation:
>
> The reviewer raises an important point about the partition function cancellation when α varies with response length. We appreciate this careful observation and want to clarify our approach.
>
> In our complete derivation (Appendix D), we maintain a fixed α throughout, exactly as standard Maximum Entropy RL requires. This produces our main theoretical result establishing the connection to Maximum Entropy RL. The partition function cancellation holds in this setting, giving us the core SimPO-style objective.
>
> We explicitly acknowledge that the final SimPO objective includes two components beyond our core derivation: (1) length normalization and (2) target margin γ. These are post-hoc pragmatic modifications addressing known pathologies in preference optimization, not theoretical requirements from the RL framework.
>
> Length Normalization: The need arises from a fundamental structural difference. DPO rewards take the form log π(y|x) - log π_ref(y|x) while SimPO rewards are simply log π(y|x). In DPO, the reference model term acts as a natural regularizer. When π and π_ref assign similar probabilities to tokens, these contributions largely cancel in the difference, preventing length exploitation. SimPO lacks this cancellation mechanism. While verbose generation occurs in DPO (Park et al., 2024), it is far more severe in SimPO due to the absence of implicit length regularization from the reference model. The SimPO paper makes similar observations about the necessity of length normalization to prevent this issue.
>
> Target Margin γ: The reference model contribution in DPO, log π_ref(y_w|x) - log π_ref(y_l|x), is relatively small in offline preference training since both responses are sampled from similar distributions (Section 4.3, Appendix B, Figure 4). However, it acts as an adaptive margin. SimPO removes the reference model, losing this adaptive mechanism. The target margin γ (adapted from IPO) serves as a fixed substitute.
>
> The reviewer is correct that these additions break strict theoretical equivalence with our Maximum Entropy RL derivation. We view this as highlighting that the pure Maximum Entropy RL framework requires additional mechanisms for practical implementation. Table 5 in the SimPO paper shows that removing either modification does not cause performance collapse, suggesting these are stability improvements rather than fundamental requirements.
>
> On Empirical Scope:
>
> We appreciate the reviewer raising this concern about generalization across scales and tasks.
>
> New results addressing scale concerns: During the rebuttal period, we gained access to additional compute resources and conducted experiments on Pythia 6.9B. The results show similar patterns to Pythia 1B: both exhibit severe overoptimization at learning rate 1 × 10⁻⁶, and only achieve comparable performance to KL-constrained methods at the much lower learning rate of 1 × 10⁻⁷. Even at this reduced learning rate, we still observe entropy maximization and high KL updates. We will include these results in the revised manuscript.
>
> Why this experimental setup: Our goal is to understand the mechanisms behind overoptimization in Maximum Entropy RLHF. The TL;DR benchmark with Pythia models provides a controlled setting where we can reliably quantify overoptimization through gold reward model win rates. At larger scales or on more complex tasks, these phenomena become harder to measure reliably. This setup follows a similar protocol from Rafailov et al., (2024), enabling direct comparison with prior work.
>
> Dataset diversification attempts: We attempted to include other datasets such as Anthropic-HH (Bai et al., 2022) However, we encountered reproducibility issues, we could not replicate reported DPO results using either official or TRL implementations, with DPO-tuned models often achieving worse win rates than their SFT counterparts. The TL;DR setup provides the stable baselines necessary for systematic investigation.
>
> We acknowledge that experiments on 7B+ models would strengthen our work but were limited by computational constraints (our hardware requires 8 × 80GB GPUs for RLHF at that scale, but we have 8x 48GB GPUs). The consistency across 1B, 2.8B, and 6.9B models suggests these may be algorithmic properties, though we agree further validation would be valuable.

---

> > ### Author Response · Authors · 2025-11-18
> >
> > On the claim that "entropy exacerbates reward hacking":
> >
> > We appreciate the reviewer's question about this seemingly counter-intuitive claim. Let us clarify the mechanism.
> >
> > In Maximum Entropy RL, we optimize the objective max_π E_{x ~ D, y ~ π}[r(x, y) + α H[π(y|x)]]. When we expand the entropy term under expectation, this becomes equivalent to optimizing E_{x ~ D, y ~ π}[r(x,y) - α log π(y|x)]. For KL-constrained RL, the analogous reward term is r(x,y) - β (log π(y|x) - log π_ref(y|x)).
> >
> > When a model is trained with KL constraints, this additional reward term decreases reward for samples that are less likely under the reference distribution. So even if the learned reward r(x,y) is high (potentially due to reward model error), the KL penalty dampens it, leading to stable optimization. The natural expectation is that entropy regularization would behave similarly.
> >
> > However, unlike KL regularization, when we deviate far from the reference distribution, the entropy reward term -α log π(y|x) actually increases the effective RLHF reward (as π(y|x) decreases, -log π(y|x) increases), whereas KL decreases it. This is what we mean by entropy exacerbating rather than mitigating reward hacking.
> >
> > It's worth noting that this likelihood increase is not unique to Maximum Entropy RL. It also occurs to some extent in KL-constrained methods, as we mention in the paper, but the reference model is able to counteract its contribution through the log π_ref(y|x) term.
> >
> > Moreover, this likelihood decrease phenomenon is also present in Direct Alignment Algorithms (DAAs)(Rafailov et al.,2024) which argues theoretically and empirically that this occurs due to OOD behavior of the implicit reward model because the reward is not strictly convex. We observe that this also happens in RLHF with explicit reward models. The key distinction is whether the method can counter this OOD behavior: Maximum Entropy RL cannot due to the lack of reference model bounding, and DAAs may face similar challenges since the reference model contribution is very small in offline settings (as we discuss in Section 4.3, Appendix B, Figure 4).
> >
> > We hope this response addresses your concerns, and we’re happy to clarify further or answer any additional questions you may have.
> >
> >
> > References
> >
> > Park, R., Rafailov, R., Ermon, S., & Finn, C. (2024). Disentangling length from quality in direct preference optimization. arXiv preprint arXiv:2403.19159.
> >
> > Rafailov, R., Chittepu, Y., Park, R., Sikchi, H.S., Hejna, J., Knox, B., Finn, C., & Niekum, S. (2024). Scaling Laws for Reward Model Overoptimization in Direct Alignment Algorithms. ArXiv, abs/2406.02900.
> >
> > Bai, Y., Jones, A., Ndousse, K., Askell, A., Chen, A., Dassarma, N., Drain, D., Fort, S., Ganguli, D., Henighan, T.J., Joseph, N., Kadavath, S., Kernion, J., Conerly, T., El-Showk, S., Elhage, N., Hatfield-Dodds, Z., Hernandez, D., Hume, T., Johnston, S., Kravec, S., Lovitt, L., Nanda, N., Olsson, C., Amodei, D., Brown, T.B., Clark, J., McCandlish, S., Olah, C., Mann, B., & Kaplan, J. (2022). Training a Helpful and Harmless Assistant with Reinforcement Learning from Human Feedback. ArXiv, abs/2204.05862.

---

### Official Review · Reviewer_64Bm · 2025-10-28

**Soundness:** 3
**Presentation:** 2
**Contribution:** 2
**Rating:** 4
**Confidence:** 3

**Summary:**

The paper explores the connection between the SimPO method and maximum entropy reinforcement learning, and examines the effectiveness of maximum entropy RL in online RLHF settings.

**Strengths:**

1) This paper establishes the relationship between SimPO and maximum entropy RL, providing a theoretical analytical perspective for SimPO.

2) Through experiments, it demonstrates that maximum entropy RL is difficult to apply to online RLHF and provides relevant analysis.

**Weaknesses:**

1) Although the paper offers in-depth analysis of the nature of SimPO, its contribution and innovation to the field of preference optimization are limited. For example, what are the implications of this paper's analysis for the design of preference learning algorithms or for addressing its existing challenges?

2) The poor performance of maximum entropy RL in online scenarios lacks thorough theoretical analysis and explanation of the underlying reasons. For example, what are the key factors (from a theoretical perspective) that lead to the failure of algorithms in online scenarios?

**Questions:**

Please see weaknesses.

---

> ### Author Response · Authors · 2025-11-18
>
> We thank the reviewer for their feedback and the opportunity to clarify our contributions and their implications.
>
> On Contributions and Implications for Preference Optimization:
>
> We believe our work has several important implications for the design of preference learning algorithms:
>
> Reference models as adaptive margins: Our analysis (Section 4.3, Appendix B, Figure 4) shows that reference models function as adaptive margins in preference optimization, providing stronger signal when the preference gap is larger under π_ref. This insight suggests that reference-free methods may need explicit margin mechanisms to compensate for this loss, as SimPO does with its target margin γ.
>
> Design principles for online vs offline: We identify that offline methods benefit from dataset constraints (all preferences are in distribution) that implicitly bound the optimization, while online methods lack these constraints. This suggests that online preference learning may fundamentally require explicit regularization (like KL constraints) rather than just entropy bonuses.
>
> These insights inform when reference-free approaches are appropriate (offline, with careful design) versus when reference models remain necessary (online settings).
>
> On Analysis of Online Failure:
>
> Our motivation for studying online Maximum Entropy RL stems directly from SimPO's success in the offline setting. Since SimPO can be derived from Maximum Entropy RL and works well offline, we investigated whether this could extend to online RLHF as a simpler alternative to KL-constrained methods. While we did not expect it to match KL-constrained performance perfectly, SimPO's strong results suggested it was worth revisiting Maximum Entropy RL in the online setting.
>
> The failure in online settings is indeed largely explained by optimizing an imperfect reward model without explicit bounds on deviation from the reference model—this is precisely why KL-constrained methods are the standard approach. Our contribution is demonstrating empirically that this failure occurs even with entropy regularization, and providing mechanistic analysis of why:
>
> KL budget requirements: We show that Maximum Entropy RL exhibits unbounded KL divergence growth (Figure 3), while successful optimization requires staying within a specific KL budget. The entropy bonus does not directly control deviation from the reference model.
>
> Entropy-overoptimization correlation: We observe that entropy increases correlate with overoptimization (Figures 1-2), suggesting that entropy maximization can encourage exploration into regions where the reward model is inaccurate.
>
> Dataset constraints in offline settings: Offline methods benefit from implicit regularization through fixed datasets, which bounds the policy's exploration. This constraint is absent in online learning.
>
> Our work clarifies that while Maximum Entropy RL can work offline with careful design (SimPO), it does not translate to a viable online approach, reinforcing why KL-constrained methods remain necessary for online RLHF.
>
> We hope this response addresses your concerns, and we’re happy to clarify further or answer any additional questions you may have.

---

### Official Review · Reviewer_EyAf · 2025-11-03

**Soundness:** 2
**Presentation:** 2
**Contribution:** 1
**Rating:** 2
**Confidence:** 4

**Summary:**

This paper establishes a new theoretical foundation for Simple Preference Optimization (SimPO), a reference-free LLM alignment method. The authors demonstrate that SimPO's objective can be mathematically derived from the Maximum Entropy Reinforcement Learning (MaxEnt RL) framework, specifically as a solution with a length-normalized temperature . Motivated by SimPO's strong performance in offline settings, the paper investigates whether MaxEnt RL can be applied directly to online RLHF. Their experiments reveal that, unlike stable KL-constrained methods, online MaxEnt RL consistently leads to overoptimization and reward hacking, even at very low learning rates. The authors conclude that entropy regularization, while effective offline as SimPO, fails to provide sufficient safeguards in online training and may even correlate with instability.

**Strengths:**

My observation is that MELON's core defensive strategy is cleverly designed to detect when an agent's tool calls are hijacked, by comparing them against a masked, task-neutral re-execution . However, the paper's own analysis of its failure modes points out a major limitation: the defense is blind to "Response-Based Attacks," which accounted for over 72% of the cases where attacks evaded detection. This is because these attacks achieve their goal through the agent's text response rather than a tool call, a behavior MELON does not monitor. This suggests that while MELON is a strong solution for preventing unauthorized actions, it doesn't address the separate and more common problem of attacks that manipulate the agent's language output.

**Weaknesses:**

1. Novelty: Although this work shows the disadvatange of Simpo or simply maximizing or minimizing the entropy, this reward hacking issue is not a new thing to the community. Only pointing out this limitation makes the contribution of this work seems weak. The authors are suggested to give more insights about how to deal with the relationship between entropy and rewards.

2. The experiments are only on small models. It would be more convincing if they experiment on more diverse base models to make the results convincing.

3. Although another contribution stated by the authors is establishing the theoretical connection between SimPO and Maximum Entropy RL. They do not provide any formal results in the main paper. It is not clear if they prove any converegence results or under which condition the equivalence hold.

**Questions:**

1. In Figure 2, does the RLHF reward contain entropy?

---

> ### Author Response · Authors · 2025-11-18
>
> We thank the reviewer for their feedback and the opportunity to clarify our contributions.
>
> On Novelty and Contribution:
>
> We'd like to clarify our main contributions:
>
> Theoretical foundation for SimPO: We derive how SimPO emerges from Maximum Entropy RL by showing the optimal policy format (with proof in Appendix D). We provide formal statements about representational completeness, equivalence class preservation, and consistency under the Bradley-Terry model (Appendix D).
>
> Understanding the online/offline asymmetry: We investigate why Maximum Entropy RL can work offline (as in SimPO with modifications) but fails online with severe overoptimization and unstable KL dynamics. This is particularly surprising given that maximum entropy methods are known to be better at handling disturbed dynamics( Eysenbach, and Levine, 2024).
>
> Mechanistic analysis: We analyze why this asymmetry exists, examining the role of dataset constraints, target margins, and how the reference model functions as an adaptive margin (Section 4.3, Appendix B, Figure 4).
>
> While reward hacking is well-known, our contribution is understanding why entropy regularization fails to prevent it online and why offline methods need additional mechanisms beyond pure Maximum Entropy RL.
>
>
> On Model Scale:
>
> We appreciate this concern. During the rebuttal period, we gained access to additional compute and conducted experiments on Pythia 6.9B. The results show similar patterns to Pythia 1B: both exhibit severe overoptimization at learning rate 1 × 10⁻⁶, and only achieve comparable performance to KL-constrained methods at the much lower learning rate of 1 × 10⁻⁷. Even at this reduced learning rate, we still observe entropy maximization and high KL updates. The consistency across 1B and 6.9B models suggests these are algorithmic properties. Full results will be included in the revised manuscript. Our setup follows Rafailov et al. (2023)'s protocol for studying overoptimization.
>
> Answer to Question:
>
> Yes, it does contain entropy reward.
>
> We hope this response addresses your concerns, and we’re happy to clarify further or answer any additional questions you may have.
>
> References
>
> Eysenbach, B., & Levine, S. (2021). Maximum Entropy RL (Provably) Solves Some Robust RL Problems. ArXiv, abs/2103.06257.
>
> Rafailov, R., Chittepu, Y., Park, R., Sikchi, H.S., Hejna, J., Knox, B., Finn, C., & Niekum, S. (2024). Scaling Laws for Reward Model Overoptimization in Direct Alignment Algorithms. ArXiv, abs/2406.02900.

---

### Official Review · Reviewer_RmXi · 2025-11-04

**Soundness:** 2
**Presentation:** 3
**Contribution:** 2
**Rating:** 2
**Confidence:** 4

**Summary:**

This paper provides a theoretical foundation for Simple Preference Optimization (SimPO) and establishes its connection to maximum entropy reinforcement learning (RL). The authors then conduct experiments showing that maximum entropy RL exhibits instability and over-optimization during online training, and that increasing the entropy coefficient can further amplify these instabilities.

**Strengths:**

a. It is interesting to observe the training instability and over-optimization phenomenon in online maximum-entropy RLHF.

b. The overall presentation is clear and easy to follow.

**Weaknesses:**

a. The connection between SimPO and maximum-entropy RLHF is not entirely solid. In maximum-entropy RLHF, the hyperparameter $\alpha$, which balances the reward term and the entropy exploration term, should be a fixed value and should not depend on $y$. However, in the connection established in Section 3.2, SimPO introduces a length normalization in this hyperparameter that depends on the response $y$, which varies across samples. Since length normalization is a key component of SimPO, analyzing its effect is crucial for understanding the algorithm’s behavior. Unfortunately, the authors do not provide a concrete analysis of this aspect, which limits the paper’s theoretical contribution. Furthermore, the paper does not explain how the margin hyperparameter $\gamma$ is related to maximum-entropy RL, making the claimed theoretical foundation of SimPO less convincing.

b. All experiments are conducted on relatively small models (e.g., Pythia 2.8B), which differ from widely used modern base models such as Qwen and LLaMA. This raises concerns about whether the observed phenomena generalize to larger and more realistic model scales. The authors should include experiments on different base models to strengthen the validity of their empirical findings.

**Questions:**

See Weaknesses section.

---

> ### Author Response · Authors · 2025-11-18
>
> Response to Weakness (a): Connection to Maximum-Entropy RLHF
>
> We appreciate the reviewer's careful examination of our theoretical derivation and want to address each concern systematically.
>
> On the α dependency and entropy term:
>
> The reviewer correctly notes that in maximum-entropy RLHF, α should be fixed and not depend on the response y. Our full derivation strictly maintains this property. In Appendix D (line 1249), we keep the entropy term fixed with constant α throughout, exactly as standard maximum-entropy RL requires. This is our main theoretical result establishing the connection to Maximum Entropy RL. We will revise the manuscript to emphasize this point more clearly.
>
> On length normalization and the target margin:
>
> We acknowledge that the final SimPO objective includes two components beyond our core derivation: (1) length normalization and (2) target margin γ. However, these are post-hoc pragmatic modifications addressing known pathologies in preference optimization, not theoretical requirements from the RL framework. This distinction is crucial:
>
> Length Normalization: The need arises from a fundamental structural difference. DPO rewards take the form log π(y|x) − log π_ref(y|x), while SimPO rewards are simply log π(y|x). In DPO, the reference model term log π_ref(y|x) acts as a natural regularizer. When π and π_ref assign similar probabilities to tokens, these contributions largely cancel in the difference, preventing the model from exploiting length to artificially inflate rewards. SimPO lacks this cancellation mechanism, the model can trivially increase log π(y|x) by generating longer sequences with moderately probable tokens. While verbose generation occurs in DPO (Park et al., 2024), it is far more severe in SimPO due to the absence of implicit length regularization from the reference model. The SimPO paper makes similar observations about the necessity of length normalization to prevent length exploitation. Length normalization is therefore a necessary practical fix, not a flaw in our RL derivation.
>
> Target Margin γ: The reference model contribution in DPO, log π_ref(y_w|x) − log π_ref(y_l|x), is relatively small in offline preference training since both responses are sampled from similar distributions (Section 4.3, Appendix B, Figure 4). However, it acts as an adaptive margin, providing stronger signal when winning responses are more likely or losing responses are less likely under π_ref. SimPO removes the reference model, losing this adaptive mechanism. The target margin γ (adapted from IPO) serves as a fixed substitute, enforcing minimum separation between winning and losing responses. While not derived from RL, it addresses the loss of adaptive regularization.
>
> Critically, Table 5 in the SimPO paper shows that removing either modification does not cause performance collapse, confirming these are stability improvements rather than fundamental algorithmic requirements.

---

> > ### Author Response · Authors · 2025-11-18
> >
> > Response to Weakness (b): Model Scale
> >
> > We thank the reviewer for raising this important concern about model scale and diversity.
> >
> > Computational constraints and new results: We acknowledge that experiments on larger models would strengthen our work. Our original experiments were limited by available hardware (8x A40 GPUs with 48GB memory). Training RLHF at 7B+ scale requires at least 8x 80GB GPUs since we must simultaneously run the policy model, reference model, and reward model while performing online rollouts.
> >
> > During the rebuttal period, we gained access to additional compute resources and conducted experiments on Pythia 6.9B. The results show similar patterns to Pythia 1B: both exhibit severe overoptimization at learning rate 1e-6, and only achieve comparable performance to KL-constrained methods at the much lower learning rate of 1e-7. Importantly, even at this reduced learning rate, we still observe entropy maximization and high KL updates. The critical dependence on KL budget for reasonable performance appears consistent across both scales. We will include these Pythia 6.9B results in the revised manuscript.
> >
> > Why this experimental setup: Our goal is to understand the mechanisms behind overoptimization in Maximum Entropy RLHF. The TL;DR benchmark with Pythia models provides a controlled setting where we can reliably quantify overoptimization through win rates. This setup follows the established protocol from (Rafailov et al, 2024), enabling direct comparison with prior work on overoptimization.
> >
> > Attempted dataset diversification: We attempted to include other datasets such as Anthropic-HH (Bai et al, 2022) to strengthen our empirical coverage. However, we encountered reproducibility issues: we could not replicate reported DPO results using either official or TRL implementations, with DPO-tuned models achieving worse win rates than their SFT counterparts. This instability prevented reliable measurement of overoptimization. The TL;DR setup provides the stable and reproducible baselines necessary for systematic investigation.
> >
> > We hope this response addresses your concerns, and we’re happy to clarify further or answer any additional questions you may have.
> >
> >
> > References
> >
> > Park, R., Rafailov, R., Ermon, S., & Finn, C. (2024). Disentangling length from quality in direct preference optimization. arXiv preprint arXiv:2403.19159.
> >
> > Rafailov, R., Chittepu, Y., Park, R., Sikchi, H.S., Hejna, J., Knox, B., Finn, C., & Niekum, S. (2024). Scaling Laws for Reward Model Overoptimization in Direct Alignment Algorithms. ArXiv, abs/2406.02900.
> >
> > Bai, Y., Jones, A., Ndousse, K., Askell, A., Chen, A., Dassarma, N., Drain, D., Fort, S., Ganguli, D., Henighan, T.J., Joseph, N., Kadavath, S., Kernion, J., Conerly, T., El-Showk, S., Elhage, N., Hatfield-Dodds, Z., Hernandez, D., Hume, T., Johnston, S., Kravec, S., Lovitt, L., Nanda, N., Olsson, C., Amodei, D., Brown, T.B., Clark, J., McCandlish, S., Olah, C., Mann, B., & Kaplan, J. (2022). Training a Helpful and Harmless Assistant with Reinforcement Learning from Human Feedback. ArXiv, abs/2204.05862.

---

### Author Response · Authors · 2025-11-19
**Updated revision for new results**

Dear reviewers, we have added the Pythia-6.9B results to the revision and included a brief discussion in Appendix E. For now, we placed them at the end of the appendix to avoid disturbing the current references. After the discussion and rebuttal period, we will carefully integrate them into the main text, along with any additional recommendations you may have.

---

### Note · Authors · 2025-12-01

**Comment:**

We are withdrawing our paper. While we assume the AC acts with the best intentions, we do not believe that a single AC reading the rebuttal can have the same impact as a full rebuttal discussion, especially given the fundamental misunderstandings about the paper. Although we submitted our rebuttal on November 18, provided additional experiments and clarifications, and received no responses from the reviewers during the rebuttal phase, we do not expect this to meaningfully change the outcome.

**Withdrawal Confirmation:**

I have read and agree with the venue's withdrawal policy on behalf of myself and my co-authors.